# Adverse event prediction using a task-specific generative model

**Otto Lönnroth** [1]  **Siddharth Ramchandran** [1]  **Pekka Tiikkainen** [2]  **Mine Öğretir** [1]  **Jussi Leinonen** [2]
**Harri Lähdesmäki** [1]

## Abstract

Longitudinal data analysis is essential in various fields, providing insights into associations between interpretable explanatory variables and temporal response variables. Recent progress in generative modelling has demonstrated models that can learn low-dimensional representations of complex longitudinal data and capture intricate interactions between high-dimensional features. Ideally, the trained generative model can be used for various downstream tasks, such as data generation, prediction and classification. In this work, we evaluate the performance of the longitudinal variational autoencoder model in predicting adverse events in clinical trials. We also propose a general training approach that can learn versatile generative models while simultaneously optimising performance on a specific downstream task. Our experiments on two simulated datasets and one clinical trial dataset demonstrate that the proposed training objective provides results that are either comparable or better than results obtained with the standard training methods. Our results also suggest that longitudinal information is useful for adverse event prediction in clinical trials.

## 1. Introduction

Longitudinal data arises in numerous fields, including psychology, sociology, economics, medicine, public health, etc. and involves repeated measurements over time for each subject. Due to its temporal nature, such data provides valuable insights into the interrelationships among explanatory variables and response variables, enabling the discovery of temporal associations and putative causal relationships. Furthermore, longitudinal data is often high dimensional, contains missing values, and has both independent explanatory covariates and dependent response variables.

Variational autoencoders (VAE) (Kingma & Welling, 2014; Rezende et al., 2014) have become a popular method to learn low-dimensional representations of complex data, facilitating data compression and reconstruction as well as generation of new data samples. Conditional variational autoencoders (CVAE) (Sohn et al., 2015) extend the standard VAEs by incorporating interpretable auxiliary covariate information directly in the generative and inference models. A limitation of both standard VAEs and CVAEs is that they assume the independence of data points, thus failing to capture, e.g., instance-specific (or patient-specific) temporal structure or correlations across all the samples.

Gaussian process prior VAEs (GP-VAE) and its extensions (Casale et al., 2018; Fortuin et al., 2020; Ramchandran et al., 2021) have been proposed to model arbitrary correlations across samples. In GP-VAEs, the i.i.d. standard Gaussian prior is replaced with a Gaussian process (GP) prior [see Williams & Rasmussen (2006) for an introduction to GPs]. Building upon this, the longitudinal VAE (L-VAE) (Ramchandran et al., 2021) uses a multi-output additive GP prior that captures both shared as well as patient-specific temporal structures by utilising the interpretable auxiliary covariate information. Therefore, L-VAE is a generative model that is specifically designed for longitudinal data.

Adverse events (also known as adverse reactions or adverse effects) refer to any undesirable or harmful events that occur during the course of a drug trial or after the administration of a medication. These events may range from mild discomfort to severe, life-threatening complications. Adverse events can be caused by the drug under investigation, concomitant medication, the underlying condition being treated, or other factors related to the trial process. Adverse events are monitored and recorded to evaluate the safety of the treatment; however, they are frequently under-reported. Therefore, predicting these events can help in assessing and understanding the potential risks associated with a medication before it can be used in the general population.

**Contribution:** In this work, we evaluate the performance of longitudinal latent variable models (L-VAE in particular) for the task of predicting adverse events in clinical trials. We

---

[1]Department of Computer Science, Aalto University, Espoo, Finland [2]Bayer Oy, Espoo, Finland. Correspondence to: Otto Lönnroth <otto.lonnroth@aalto.fi>.

*Workshop on Interpretable ML in Healthcare at International Conference on Machine Learning (ICML)*, Honolulu, Hawaii, USA. 2023. Copyright 2023 by the author(s).

also propose a general training approach, called D-ELBO, for deep latent variable models that combines the standard training objective with an objective that is specific to a given downstream task.

## 2. Methods

**Notation:** In our setting, we have $N = \sum_{p=1}^{P} n_p$ observations, where $P$ is the number of unique instances (i.e., patients) and $n_p$ is the number of longitudinal samples from instance $p$. The longitudinal samples are divided into two categories, $X_p$ and $Y_p$. $X_p$ contains the auxiliary covariate information for the instance $p$ as $X_p = [\mathbf{x}_1^p, \ldots, \mathbf{x}_{n_p}^p]$, where $\mathbf{x}_t^p \in \mathcal{X} = \mathcal{X}_1 \times \ldots \times \mathcal{X}_Q$ are covariates for a sample. $Q$ is the number of auxiliary covariates and $\mathcal{X}_q$ is the domain of the $q$-th covariate. $Y_p = [\mathbf{y}_1^p, ..., \mathbf{y}_{n_p}^p]$ contains longitudinal samples for instance $p$, where $\mathbf{y}_t^p \in \mathcal{Y} = \mathbb{R}^D$, and $D$ is the dimension of the longitudinal samples. Instance-specific observations form our longitudinal data $[X, Y]$, where $X = [X_1, \ldots, X_P] = [\mathbf{x}_1, \ldots, \mathbf{x}_N]$ and $Y = [Y_1, \ldots, Y_P] = [\mathbf{y}_1, \ldots, \mathbf{y}_N]$. The low-dimensional latent space is denoted as $\mathcal{Z} = \mathbb{R}^L$, where $Z = [\mathbf{z}_1, \ldots, \mathbf{z}_N] \in \mathbb{R}^{N \times L}$ and $\mathbf{z}_n$ is an embedding to dimension $L$ for sample $[\mathbf{x}_n, \mathbf{y}_n]$. For example, in the context of clinical trial data, $X$ could contain patient-specific demographics, concomitant medication and adverse events, and $Y$ could contain observed laboratory measurements.

### 2.1. Conditional variational autoencoder

CVAE is an extension to the standard VAE where the generative model is conditioned with auxiliary covariates. In general, the conditional distribution of the CVAE can be written, e.g., as $p_w(\mathbf{y}, \mathbf{z}|\mathbf{x}) = p_\psi(\mathbf{y}|\mathbf{z})p_\theta(\mathbf{z}|\mathbf{x})$. CVAE models are typically trained by maximising the evidence lower bound (ELBO), which can be expressed for a single data sample as $L(\phi, \psi, \theta, \mathbf{y}|\mathbf{x}) = E_{q_\phi}[\log p_\psi(\mathbf{y}|\mathbf{z})] - \mathrm{KL}[q_\phi(\mathbf{z}|\mathbf{y}, \mathbf{x})||p_\theta(\mathbf{z}|\mathbf{x})]$, where the expectation is over the latent variable $z$ and $q_\phi(\mathbf{z}|\mathbf{y}, \mathbf{x})$ denotes the variational approximation of the true posterior $p_w(\mathbf{z}|\mathbf{y}, \mathbf{x})$. Due to the independence of the prior $p_\theta(\mathbf{z}|\mathbf{x})$ across the data samples, optimisation of the model is straightforward to do with mini-batch based stochastic gradient descent (SGD).

### 2.2. Longitudinal variational autoencoder

The longitudinal variational autoencoder (L-VAE) is a conditional generative model as the generative process depends on auxiliary covariates. It makes use of a multi-output GP prior over the latent space $\mathbf{z}|\mathbf{x} \sim \mathcal{GP}(\mu(\mathbf{x}), k(\mathbf{x}, \mathbf{x}'|\theta))$, where we assume zero mean ($\mu(\mathbf{x}) = 0$), and $k(\mathbf{x}, \mathbf{x}'|\theta) = \sum_{r=1}^{R} k^{(r)}(\mathbf{x}^{(r)}, \mathbf{x}'^{(r)}|\theta^{(r)})$ is the sum of $R$ positive definite cross-covariance functions (CF) – one for each of the GP components and $\theta^{(r)}$ refers to the corresponding ker-

nel parameters. The additive GP model is analogous to the well-known linear mixed models, where the function is decomposed into $R$ additive effects, such that each effect depends only on a small number (typically one or two) of auxiliary covariates $\mathbf{x}^{(r)}$. The CFs can be, e.g., the squared exponential kernel for temporal effects or for other continuous covariates, binary or categorical kernel for discrete-valued inputs (such as gender), their products (interactions), or essentially any valid kernel. The L-VAE model can be trained using the standard ELBO objective $L(\phi, \psi, \theta, Y|X)$. However, due to the GP prior $p(Z|X)$, the training objective does not factorise across the data samples and, therefore, the time complexity of the standard inference would be in the order of $\mathcal{O}(N^3)$ w.r.t. the number of training data points. We use the scalable, mini-batch compatible training method from (Ramchandran et al., 2021) that uses the inducing point approximation for GPs.

### 2.3. Downstream analysis tasks

After training a conditional generative model, we may want to generate new data samples or, alternatively, use the trained model for various downstream tasks. As an example, we consider the task of classifying samples. Assume that the $q$-th covariate is binary-valued and represents whether the data samples belongs to one of the two classes. For example, in our clinical dataset, the presence or absence of a specific adverse event for a patient $p$ at time point $t$ is encoded by the $q$-th covariate, $x_{tq}^p \in \{0, 1\}$. If the value of the $q$-th covariate is missing from a new data sample $(\mathbf{x}^*, \mathbf{y}^*)$, we can use the trained model to classify it by calculating the lower bound for the log probability of the new data sample for the two alternative events as $L_i = L(\phi, \psi, \theta, Y, \mathbf{y}^*|X, \mathbf{x}^*, x_{tq}^p = i)$, where $i \in \{0, 1\}$. The prediction probability can then be computed as:

$$P_i = \frac{\exp(L_i)}{\exp(L_{1-i}) + \exp(L_i)}. \tag{1}$$

### 2.4. A new discriminative training objective

In order to create a bespoke generative model for a predetermined downstream task, we propose a new training objective D-ELBO:

$$L = \alpha \cdot \mathrm{ELBO} + (1 - \alpha) \cdot D, \tag{2}$$

where $D$ is an objective of a specific downstream task and $\alpha \in [0, 1]$ is a scaling factor. For example, in the case of the two-class prediction task described in Section 2.3, $D$ can be defined using the log of the Bernoulli likelihood as

$$D = \sum_{p=1}^{P} \sum_{n=1}^{n_p} x_{nq}^p \log(p_{nq}^p) + (1 - x_{nq}^p) \log(1 - p_{nq}^p) \tag{3}$$

where $p_{nq}^p = P_1 = P(x_{nq}^p = 1)$ is the predicted probability obtained from the generative model (see Eq. 1), and $x_{nq}^p$ is the observed value for the auxiliary covariate that is available during the training time. In the following experiments, we replace the sum over $n$ by a random subset of time points $T_p \subset \{1, \ldots, n_p\}$, with $|T_p| = 4$. For $\alpha = 1$, D-ELBO reverts back to the standard generative model training, whereas $\alpha = 0$ results in a purely discriminative objective. In practice, $\alpha$ can be chosen using cross-validation or performance on a held-out validation dataset. Overall, the training objective in Eq. 2 provides a simple approach to re-train or fine-tune, a general purpose generative model for a variety of downstream tasks as we demonstrate below.

## 3. Experiments

We demonstrate the ability of a GP prior based conditional generative model to predict adverse events with three distinct datasets: two types of simulated datasets (based on MNIST digits) and a real clinical trial dataset. The L-VAE model is learnt on the training data using the ELBO objective (**L-VAE + ELBO**) and performing adverse event prediction using Eq. 1. We also evaluate the performance of the proposed new objective function by learning the L-VAE model using the D-ELBO in Eq. 2 (**L-VAE + D-ELBO**) and then predicting using Eq. 1. We use a grid search for the $\alpha$ parameter ($\alpha \in \{0.1, \ldots, 10^{-6}, 0\}$) using the mean over the validation folds, and use the mean of the corresponding test folds as the estimate of the objective that is compared. We took the average of the results over the five independent runs and compared it against L-VAE that was trained using the standard ELBO objective (**L-VAE + ELBO**). As a baseline method, we compare our results against a standard multilayer perceptron (MLP) that is applied in a cross-sectional manner, i.e., its input data is from a single time point. We use the area under the receiver operating characteristic curve (AUROC) and the area under the precision recall curve (PR AUC) metrics to account for varying adverse event frequencies [see (Murphy, 2022)].

Different cross-covariance functions (CF) for the additive multi-output GP prior are denoted as: $f_{\text{ca}}(\cdot)$ for categorical CF, $f_{\text{se}}(\cdot)$ for the squared exponential CF, and $f_{\text{bin}}(\cdot)$ for binary CF. The definitions for different cross-covariance functions can be found in the Supplementary Material in Section B. In all the experiments, the first four time points from validation and test subjects were included in the training data. These samples were neither used for the validation nor the final evaluation.

### 3.1. Simulated dataset with MNIST digits

We created two simulated datasets that make use of MNIST digits (LeCun et al., 2010) to represent simulated biomedical longitudinal data. Both datasets consist of $P = 400$

unique digits (i.e., instances or patients) with $n_p = 20$ observations (therefore, $N = 8000$). Each instance has $Q = 4$ auxiliary covariates: unique id (*id*), gender (*gender*), time (*time*), and adverse event (*ae*). We assume two genders (for convenience) such that a number '3' represents female and '6' represents male. To simulate a shared time-dependent effect, all digit instances where shifted towards the bottom-right corner over time. In the two datasets, we simulate the adverse events differently: (1) the intensity of a digit diminishes when the event occurs, or (2) the orientation of a digit changes when the event occurs (see Supplementary Fig. 2 for an illustration). Adverse events occur in $50\%$ of the unique instances (i.e., 200 out of 400 unique instance) with the adverse event occuring in $20\%$ of the observations. In all of the experiments, the following additive multi-output GP model was used: $f_{\text{ca}}(id) + f_{\text{se}}(time) + f_{\text{ca}}(gender) + f_{\text{bin}}(ae)$.

Table 1 shows that the proposed training objective (**L-VAE + D-ELBO**) results in an improved adverse event prediction accuracy in almost all variants.

### 3.2. Clinical trial data

We demonstrate the efficacy of the two objectives for adverse event prediction using data from a randomised controlled trial (RCT) for the treatment of colorectal cancer (dataset identifier: Colorec_SanfiU_2007_131) from the open data sharing platform, 'Project datasphere' (Green et al., 2015). The control arm of the study consists of 610 subjects. We performed the following pre-processing steps:

- Laboratory measurements (*lb*), adverse event information (*ae*), vital signs (*vs*), concomitant medication (*cm*), and demographic information (*dm*) were selected as source domains.

- Information on the domains were formulated as longitudinal samples $Y$ comprising [*lb*, *vs*] and auxiliary covariates $X$ comprising [*dm*, *ae*, *cm*]. We considered only measurement time points in *ae* and *cm* where we know both the start and end time of the event.

- Time points with less than $70\%$ of information in $Y$ were excluded.

- We included the 10 most common *ae* and *cm* in $X$

- Patients with less than 5 observations were excluded.

- The measurements ($Y$) were standardised per feature.

After preprocessing, the dataset comprised $P = 480$ subjects, with $N = 6605$ observations. The dimension of the longitudinal samples in $Y$ was 30, which consists of 27 laboratory measurements and 3 vital signs. The auxiliary covariates ($X$) have a dimension ($Q$) of 24 and include: three types of demographic information (*id*, *gender*, *age*), time of

*Table 1.* Adverse event prediction accuracies on the simulated datasets for the L-VAE model, trained using the ELBO (**L-VAE + ELBO**) and the proposed D-ELBO (**L-VAE + D-ELBO**). AUROC and PR AUC values are computed on test subjects. **Higher values are better**.

| Experiment | AUROC (↑) | | PR AUC (↑) | |
|---|---|---|---|---|
| | L-VAE + ELBO | L-VAE + D-ELBO | L-VAE + ELBO | L-VAE + D-ELBO |
| Intensity fades by 5% | 0.779±0.035 | **0.847±0.030** | 0.290±0.029 | **0.449±0.063** |
| Intensity fades by 10% | 0.922±0.018 | **0.963±0.024** | 0.532±0.072 | **0.788±0.106** |
| Intensity fades by 20% | 0.998±0.001 | **0.999±0.001** | 0.986±0.009 | **0.991±0.008** |
| Rotation changes 1° | 0.791±0.017 | **0.796±0.021** | **0.244±0.021** | **0.244±0.041** |
| Rotation changes 3° | 0.863±0.016 | **0.884±0.016** | 0.595±0.054 | **0.674±0.042** |
| Rotation changes 5° | 0.945±0.029 | **0.946±0.016** | **0.789±0.047** | 0.782±0.032 |

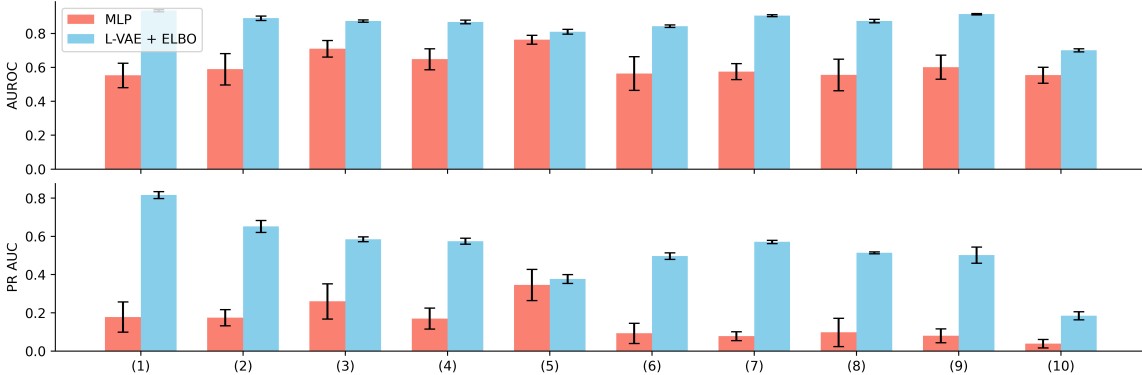

*Figure 1.* Comparison of L-VAE (**L-VAE + ELBO**) with the baseline cross-sectional approach using a MLP on the Project Datasphere dataset. The numbers on the x-axis correspond to the adverse events in Table 2. **Higher values are better**

*Table 2.* Adverse event prediction accuracies on the Project Datasphere dataset. We demonstrate the results of L-VAE with the two objectives: **L-VAE + ELBO** and **L-VAE + D-ELBO**. The occurrence (*Occ.*) is the fraction of instances with a particular adverse event, and *Mo* is the mean of the fraction of observations with a particular adverse event on the given $P$ instances. AUROC and PR AUC values are computed using 5-fold cross-validation. **Higher values are better**.

| Adverse event | Occ. (%) | Mo (%) | AUROC (↑) | | PR AUC (↑) | |
|---|---|---|---|---|---|---|
| | | | L-VAE + ELBO | L-VAE + D-ELBO | L-VAE + ELBO | L-VAE + D-ELBO |
| (1) Skin appendage conditions | 28 | 45 | 0.934±0.006 | **0.946±0.003** | 0.816±0.018 | **0.834±0.005** |
| (2) General system disorders nec | 35 | 35 | 0.890±0.013 | **0.900±0.006** | 0.652±0.031 | **0.684±0.014** |
| (3) Gastrointestinal signs and symptoms | 42 | 31 | **0.873±0.007** | 0.871±0.005 | 0.585±0.013 | **0.589±0.004** |
| (4) Gastrointestinal motility and def. cond. | 35 | 27 | 0.868±0.011 | **0.874±0.012** | **0.575±0.016** | 0.564±0.013 |
| (5) White blood cell disorders | 37 | 25 | 0.810±0.014 | **0.816±0.008** | 0.377±0.023 | **0.395±0.003** |
| (6) Oral soft tissue conditions | 22 | 25 | 0.843±0.007 | **0.857±0.007** | 0.497±0.017 | **0.500±0.006** |
| (7) Neurological disorders nec | 14 | 30 | 0.905±0.006 | **0.914±0.004** | **0.571±0.009** | 0.569±0.021 |
| (8) Respiratory disorders nec | 18 | 22 | 0.873±0.010 | **0.887±0.004** | 0.514±0.005 | **0.519±0.011** |
| (9) Appetite and general nutritional disorders | 16 | 27 | **0.914±0.004** | 0.913±0.012 | 0.502±0.042 | **0.556±0.018** |
| (10) Infections pathogen unspecified | 19 | 15 | 0.701±0.009 | **0.710±0.006** | 0.185±0.021 | **0.188±0.021** |

the measurement (*time*), 10 adverse events (*ae*), and 10 concomitant medications (*cm*). In the experiments, the structure of the GP model was: $f_{\text{se}}(time) + f_{\text{ca}}(id) + f_{\text{ca}}(gender) + \sum_{i=1}^{10} f_{\text{bin}}(ae_i) + \sum_{i=1}^{10} f_{\text{bin}}(cm_i)$.

The results of predicting 10 different adverse events in Fig. 1 show that using the L-VAE model to predict adverse events

provides a significant performance improvement over a standard cross-sectional approach using a MLP. Furthermore, Table 2 demonstrates that the proposed D-ELBO training objective (**L-VAE + D-ELBO**) improves the prediction accuracy for almost all considered adverse events.

## 4. Conclusion

In this work, we examined the applicability of a GP prior based conditional generative model to predict adverse events. Additionally, we introduced the D-ELBO, a novel objective function, designed to enhance the performance of generative models on a predetermined downstream task, while preserving its comprehensive properties. We performed evaluations on two simulated datasets and one real-world dataset. The results showed evidence of the L-VAE's capability to confidently predict adverse events across the majority of the endpoints. Furthermore, the results demonstrated the benefit of the new training objective, D-ELBO, which provided consistent results that were either comparable or better than the results obtained with the standard training objective.

## 5. Acknowledgements

We would like to acknowledge the computational resources provided by Aalto Science-IT, Finland. This work was supported by the Academy of Finland [328401] and Bayer Oy.

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

# A. Supplementary images

## A.1. Health MNIST data

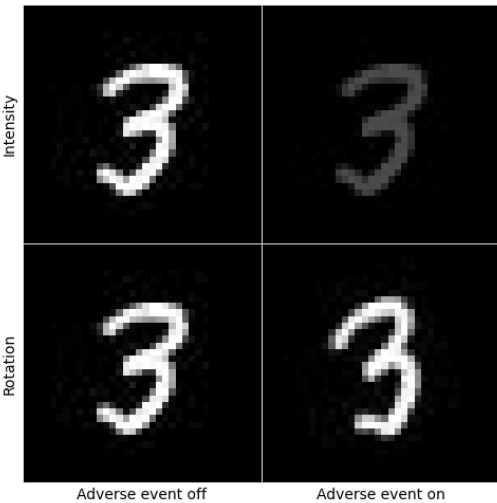

*Figure 2.* Visualisation of a simulated adverse event effect on the MNIST digits.

## A.2. Datasphere data

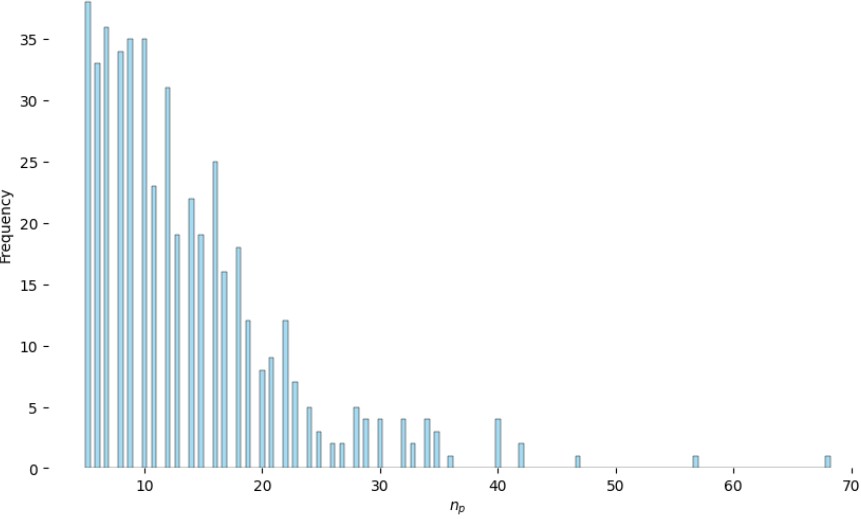

*Figure 3.* Visualisation of $n_p$ distribution on Datasphere data.

# B. Cross-covariance functions (CF)

Definitions for the cross-covariance functions used in our experiments:

**Squared exponential CF**

$$k_{\text{se}}(\mathbf{x}^{(r)}, \mathbf{x}'^{(r)}|\theta_{\text{se}}) = \sigma_{\text{se}}^2 \exp\left(-\frac{(x-x')^2}{2l_{\text{se}}^2}\right), \quad \theta_{\text{se}} = (\sigma_{\text{se}}^2, l_{\text{se}}),$$

where $\mathbf{x}^{(r)} = x \in X_j$ denotes a univariate continuous-valued covariate, $\sigma_{\text{se}}^2$ is the magnitude parameter (also called scale), and $l_{\text{se}}$ is the length scale. The magnitude controls the marginal variance of the GP and the length-scale controls its smoothness (Rasmussen & Williams, 2006).

**Categorical CF**

$$k_{\text{ca}}(\mathbf{x}^{(r)}, \mathbf{x}'^{(r)}) = \begin{cases} 1, & \text{if } x = x' \\ 0, & \text{otherwise} \end{cases}, \quad \theta_{\text{ca}} = \emptyset,$$

where $\mathbf{x}^{(r)} = x \in X_j$ denotes a categorical or discrete covariate.

**Binary CF**

$$k_{\text{bin}}(\mathbf{x}^{(r)}, \mathbf{x}'^{(r)}) = \begin{cases} 1, & \text{if } x = x' = 1 \\ 0, & \text{otherwise} \end{cases}, \quad \theta_{\text{bin}} = \emptyset,$$

where $\mathbf{x}^{(r)} = x \in X_j = \{0, 1\}$ denotes a binary-valued covariate.

## C. Supplementary tables

| | Hyperparameter | Value |
|---|---|---|
| | Dimensionality of input | $36 \times 36$ |
| | Number of convolution layers | 2 |
| | Number of filters per convolution layer | 144 |
| | Kernel size | $3 \times 3$ |
| | Stride | 2 |
| Inference | Pooling | Max pooling |
| network | Pooling kernel size | $2 \times 2$ |
| | Pooling stride | 2 |
| | Number of feedforward layers | 2 |
| | Width of feedforward layers | 300, 30 |
| | Dimensionality of latent space | $L$ |
| | Activation function of layers | RELU |
| | Dimensionality of input | $L$ |
| | Number of transposed convolution layers | 2 |
| | Number of filters per transposed convolution layer | 256 |
| Generative | Kernel size | $4 \times 4$ |
| network | Stride | 2 |
| | Number of feedforward layers | 2 |
| | Width of feedforward layers | 30, 300 |
| | Activation function of layers | RELU |

*Table 3.* Neural network architectures used in the MNIST digits (L-VAE).

|  | Hyperparameter | Value |
|---|---|---|
| Inference network | Dimensionality of input | 30 |
|  | Number of feedforward layers | 2 |
|  | Number of elements in each feedforward layer | $15, 7$ |
|  | Dimensionality of latent space | $L$ |
|  | Activation function of layers | RELU |
| Generative network | Dimensionality of input | $L$ |
|  | Number of feedforward layers | 2 |
|  | Number of elements in each feedforward layer | $7, 15$ |
|  | Activation function of layers | RELU, Sigmoid |

*Table 4.* Neural network architectures used in the clinical trial dataset (L-VAE).

| Hyperparameter | Value |
|---|---|
| Dimensionality of input | 54 |
| Number of feedforward layers | 2 |
| Number of elements in each feedforward layer | $15, 7$ |
| Activation function of layers | RELU, Sigmoid |

*Table 5.* Neural network architecture used in the clinical trial dataset (MLP).

