# OpenReview forum: "Adverse event prediction using a task-specific generative model"
_ICML.cc/2023/Workshop/IMLH — IMLH 2023 PosterShortPaper_

### Official Review · Reviewer_1HBV · 2023-06-10

**Rating:** 4
**Confidence:** 3

**Review:**

Summary:
This paper explores using Longitutional VAE in a medical setting, as well as proposes a new training objective to better train VAEs jointly with a supervised end task.

Strengths:
- Paper is clearly written
- Interesting use case and results seem good

Weaknesses:
- Poor evaluation. Only evaluated on one medical task, with a very weak baseline (cross-sectional MLP). Would like to see more datasets and stronger baselines such as transformers or LSTMs, or even an MLP that has access to previous data.
- Not relevant to workshop. While it is focused on medicine, I don't think this is an interpretable ML approach, or focused on measuring uncertainty etc, therefore not a good fit for this workshop.

---

### Official Review · Reviewer_VonZ · 2023-06-12
**Adverse event prediction using a task-specific generative model**

**Rating:** 7
**Confidence:** 4

**Review:**

**Summary:** The authors show the feasibility to utilize longitudinal latent variable models (e.g., L-VAE) to predict adverse events in data analysis. Specifically, they model the prior probability with Gaussian Process on the latent space of the auxiliary covariate information, and design a novel training objective (D-ELBO) with an additional task specific loss compared to ELBO. The experiment result shows that their model can effectively work on detecting adverse events on two simulated MNIST datasets and one real clinical dataset.

**Strength**
1. The paper is well written and easy to read.
2. The proposed method, L-VAE+D-ELBO, for detecting adverse events sounds novel to me.
3. The authors show comprehensive improvement compared to the baseline method.

**Weakness**
1. It would be great if the authors can add more detailed descriptions/explanation on some terms. (For example, the four cross-covariance functions in line 149 and the w, psi and theta in line 77).
2. It would be great if the authors can include some related adverse event prediction papers.
3. Question for Fig1: Is MLP the only baseline method? As I feel using MLP for comparing VAE is a bit unfair.
4. Please correct some typos, for example, “my” in line 78 should be “by”

---

### Meta-Review · Area_Chair_sWCX · 2023-06-19

**Recommendation:** Accept (Poster)
**Confidence:** 3

**Metareview:**

The study demonstrates the practicality of utilizing longitudinal latent variable models, such as L-VAE, for predicting adverse events in data analysis. The authors employ a Gaussian Process to model the prior probability on the latent space of the auxiliary covariate information and introduce a new training objective (D-ELBO) with an additional task-specific loss compared to ELBO.

The paper has been deemed interesting by all reviewers. Some concerns have been raised regarding the evalation and relevance to the workshop. The authors are encouraged to address these weaknesses in the next revision, as outlined in the reviews.

---

### Decision · Program_Chairs · 2023-06-20

Accept (Poster Short Paper)